# PRISIM: Privacy Preserving Synthetic Data Simulator

**Dr. Subhrajit Samanta** *
ZS Associates
subhrajit.samanta@zs.com

**Shantanu Chandra**
ZS Associates
shantanu.chandra@zs.com

**Dr. PKS Prakash**
ZS Associates
prakash.prakash@zs.com

**Srinivas Alva**
ZS Associates
srinivas.alva@zs.com

**Srinivas Chilukuri**
ZS Associates
srinivas.chilukuri@zs.com

## Abstract

Data sharing in a collaborative environment is instrumental to propel innovation; however, privacy can pose a serious threat when sharing data as it comes with the risk of sensitive information leakage. On the other hand, analytical utility is another key factor to consider while sharing data to ensure its usability. Therefore, this research primarily focuses on the assessment and preservation of privacy and utility within centralized tabular data which is one of the most common types of data used across industries (e.g. HR, CRM, healthcare). The state-of-the-art (SOTA) centralized privacy preservation techniques, such as statistical anonymization (using generalization, binning, suppression, etc.) and differential privacy (DP) methods focus heavily on data privacy and ignore the analytical utility to a large extent. Hence, in this paper we propose a novel synthetic data generation-based approach with a statistical distance-based privacy-preserving mechanism (the framework is referred to as PRISM) to ensure analytically useful private synthetic data. PRISIM is validated across five open-source data sets and compared against SOTA Differentially Private GANs and we observed on average $> 20\%$ higher retention of utility while maintaining a similar level of privacy.

## 1 Introduction

Data sharing among the AI community is one of the key factors that has fueled the accelerated adoption of AI/ML across industries for a wide variety of applications. However, increasing privacy awareness among practitioners has led to regulations such as General Data Protection Regulation (GDPR) in EU which is impacting data sharing among the broader AI Community consisting of sensitive information. In order to achieve privacy, traditional methods compromise on the quality of the data-set which ends up limiting its analytical utility towards building AI/ML applications.

In literature, researchers have explored various approaches such as federated learning, statistical masking and differential privacy. Federated learning-based approaches address the privacy issue by keeping data on decentralized edge devices [1, 2] whereas other two: statistical masking and differential privacy focus on attaining privacy within centralized data-sets. Statistical masking-based approaches focus on data generalization, while differential privacy-based mechanisms use noise injection to achieve required level of privacy. Both statistical masking and differential privacy-based approaches have been shown to significantly impact the analytical utility of the data-sets limiting their applications. Recently researchers have started using synthetic data as an alternative for privacy preservation as it reduces the likelihood of one-to-one mapping substantially while keeping the overall data distribution intact. However, presence of the multivariate relationship in the synthetic data (learned from real) can expose sensitive attribute information under an inference attack. To address this issue, differential privacy based synthetic data generation approaches such as DPGAN,

NeurIPS 2022 Workshop on Synthetic Data for Empowering ML Research.

[3], DTGAN [4] have come to prominence. However, the fidelity of synthetic samples from these methods are generally poor which leads to significant reduction of the analytical utility. Additionally, most of the research published in this area focus on theoretical guarantee of privacy with little to no focus on empirical evidence of privacy.

Therefore, this article focuses on mainly two themes, (a) generating private synthetic tabular data which is designed to minimize the distance-based re-identification attacks [5]; and (b) providing empirical evidence of privacy and utility from the synthetic private data to drive real world adoption.

## 2 Existing Work

Privacy can be a prominent issue due to the presence of direct or indirect identifiers. Statistical methods such as anonymization or pseudonymization focus on suppressing (or masking) such fields using standard protocols (i.e. HIPAA rules) which ends up reducing the utility of the data drastically. Researchers have also explored differential privacy (DP) [6] that ensures statistically indistinguishable output on a pair of neighboring databases thereby providing the theoretical guarantee of privacy, however DP as well suffers the same challenge of diminished utility like the statistical methods.

Synthetic data generation by learning the co-variate structure of the real data is another efficient and accepted form of privacy preservation as they can retain the characteristics of the original data without any one-on-one mapping with the original records. However, [7] showed that even with synthetic data, privacy can be a concern. To address the issue, researchers integrated DP with synthetic data generation methods such as GANs [8] to develop deferentially private GANs. Two of the most important works in this field are Differentially Private GAN (DPGAN) [3] and Private Aggregation of Teacher Ensemble GANs (PATEGAN) [7]. DPGAN trains a generator by carefully adding noise to gradients during the learning procedure to ensure DP guarantee. One major problem which is common across these methods is that they do not provide an empirical evidence of privacy and mostly rely on a subjective downstream ML task to show that with higher privacy, the ML utility decreases and therefore argue that a better privacy is achieved.

In this study, the privacy vs utility trade-off is explored with five sensitive real-world data-sets (HR, MIMIC-III, Airlines, Heart-failure and ADULT). All of these data-sets are multivariate, mixed type (continuous+categorical) and tabular in nature. Conditional Tabular GAN (CTGAN) [9] is the current SOTA for generating tabular data. It utilizes a conditional generator to generate the continuous columns conditioned on the discrete features. First, we extended the CTGAN architecture to DP-CTGAN by adding a differential privacy mechanism (similar to DPGAN) to explore the deficits of DP. The major limitation that was found with this approach is the significant reduction in analytical utility due to noise accumulation. In this study, the aim is to bridge these gaps with PRISIM.

## 3 Proposed Privacy Mechanisms

This section starts with the discussion on how we extend the vanilla CTGAN architecture to build a Differentially Private CTGAN. Next, we propose the novel Hybrid (CTGAN + distance based privacy) approach referred here as PRISIM to address the challenges of DP methods as DP-CTGAN.

### 3.1 CT-GAN with Differential Privacy (DP-CTGAN)

Differential privacy policy learns important information about a population without learning any information about individual [10]. The DP guarantee can be achieved in GANs by adding controlled noise. We extend CTGAN to DP-CTGAN by following the DP-GAN [3] framework. During training of the CTGAN discriminator, Gaussian (or Laplacian) noise is added to the gradients of the loss for the weights. Further clipping of these weights are performed. [3] has shown that following the above steps, we can ensure that the generated samples (from DP-CTGAN) are differentially guaranteed. However, a common problem with such methods which is also apparent with DP-CTGAN, is the accumulation of noise. Given noise is added at a gradient level, the total noise present in the data grows rapidly and ends up perturbing the co-variate structure of the data, leading to a significant loss in analytical and ML utility (shown in the results section).

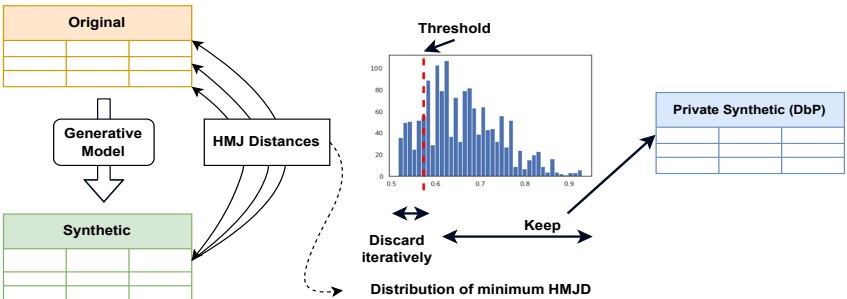

Figure 1: PRISIM Framework Illustration

## 3.2 PRISIM: CTGAN with Distance Based Privacy (DbP)

In this section, we address the challenge of diminished utility via a novel privacy mechanism that does not require any noise-injection. The proposed mechanism is a hybrid: synthetic (with CTGAN) + statistical distance based privacy (DbP) approach specifically designed to eradicate the problem of re-identification attacks. Overview of the PRISIM framework is provided in figure 1.

**Distance based Privacy:** The principal idea behind PRISIM is: if a synthetic sample is very similar to a real sample (i.e. low distance, in the original multi-dimensional feature space), then that synthetic sample is at a high risk of re-identification. Essentially, we investigate if the data synthesizer (such as CTGAN) [2] is memorizing the original samples while generating synthetic samples and thereby putting the said original sample at possible risk of re-identification. First we find the synthetic samples that are 'too-similar' to their original counterparts and tag them 'risky'.

**'Risky' Samples Identification :** To identify the risky samples, a novel heterogeneous distance metric [11] is proposed to find the distance between two tabular data points having mixed type data. In that light, Jaccard index is used as a similarity metric for categorical variables. Categorical variables (nominal) are treated as sets and the distance between two data points containing such variables can be quantified using the ratio of matches between them as shown here, $JI_{x,y} = \frac{|X \cap Y|}{X \cup Y} = \frac{|X \cap Y|}{|X|+|Y|-|X \cap Y|}$ where, $JI_{x,y}$ is the Jaccard similarity index between sets $X$ and $Y$. Set $X$ is the discrete feature values of a sample $x$ from original data-set and $Y$ is the discrete feature values of the generated sample $y$. We can further define, $JD_{x,y} = (1 - JI_{x,y})$ as the Jaccard distance. Please note Jaccard distance is always bounded between $[0, 1]$ where lower distance means high similarity and vice versa.

Distance computation in continuous space can be challenging. The limitations of the popular L1 or L2 norm distances (i.e., Manhattan, Euclidean) are that they do not take the correlations between the features into account. Therefore we argue that the Mahalanobis distance (MD) is more suited for this task as it considers the cross-covariance of the features and uses the same for scaling. For real and synthetic samples $x$ and $y$ with sample co-variance matrix (positive definite) $C$, MD can be defined as, $MD^2_{x,y} = (x-y)^T C^{-1}(x-y)$. To compute the 'sample-risk', combination of these two distances with a weighted mean squared approach is used to build the Heterogeneous Mahanalobish Jaccard Distance (HMJD) as below, $HMJD^2_{x,y} = \frac{p_1 . MD^2_{x,y} + p_2 . JD^2_{x,y}}{p_1 + p_2}$, where $p_1, p_2$ are the cardinality of continuous and categorical features respectively. Finally 'risky' sample identification is performed, by finding the closest original match (i.e. the one with the lowest HMJD) for each synthetic sample. The overall distribution of the 'lowest' HMJD is explored for all the synthetic samples (as shown in figure 1) to identify the synthetic samples for which the minimum HMJD is lower than a particular threshold e.g. the privacy threshold. Therefore, a synthetic sample $y_j$ can be identified as 'risky' when, $\arg\min_i HMJD(x_i, y_j) \le \theta$, where $x_i, y_j$ represents the real and synthetic samples and $\theta$ is the privacy threshold.

**Risk Mitigation :** The next step is the risk mitigation for the identified 'risky' synthetic samples. For this purpose, iterative removal of the 'risky' samples is performed. For example, if synthetic data-set $S$ initially had $m$ samples, then we remove the samples for which, $HMJD_j \le \theta$. Let us

---

[2]Note that we have done extensive exploration with other generative models such as VAE and Copula. Please refer to the **supplementary materials** for more details on the same.

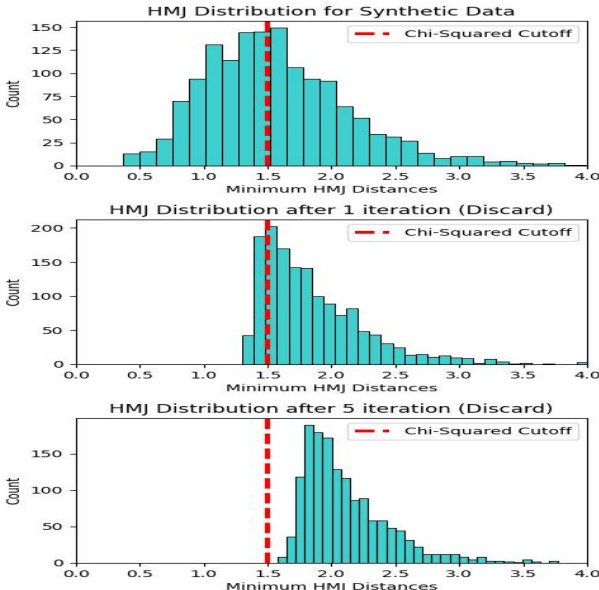

Figure 2: Shift of HMJD distribution with iterative 'risky' sample removal. It is observed that with each discard step, the distribution shifts towards right and after few iterations, the minimum HMJD becomes higher than $\theta$ for all the remaining samples i.e. maximum privacy is attained for that $\theta$.

denote the resultant data-set with $S'$. Please note that, $|S'| = |\{y_j \in S | HMJD_j > \theta\}| \leq |S|$ where $|.|$ represents the cardinality of samples in the data-set. With decreased number of samples however, the co-variate structure of the synthetic data changes along with the Mahalanobis distance (due to sample rejection). Therefore re-evaluation is needed for the HMJD for $S'$ with the remaining synthetic samples to look at the updated minimum HMJD distribution (as in the first step). This process is repeated, till no synthetic sample has a HMJD less than the privacy threshold $\theta$.

This process leads to a right shift of the HMJD distribution due to iterative removal of synthetic samples with low distances to the original (as samples with higher distances are kept to generate the private data-set). The rejection process is stopped once for all $y_j \in S'$ the $HMJD_j > \theta$ as illustrated in figure 2, thus leading to the desired privacy against distance based 're-identification'.

**The choice of threshold** $\theta$ has a significant impact on the proposed DbP mechanism. Higher value of the threshold leads to removal of more synthetic samples resulting in stricter privacy (lower possibility of re-identification) and a lower number of samples and lower utility. Whereas, with a lower $\theta$ fewer synthetic samples are discarded hence retains better utility, but the privacy of the generated data is low. This threshold should be chosen empirically by an expert user. To achieve a desired number of private samples, one should start with generating a synthetic pool of roughly 5 times of that (empirically observed). A detailed instruction on how to choose $\theta$ in a data-driven manner (chi-squared cut-off) is provided along with a sensitivity analysis in the **supplementary material**.

## 4   Privacy Evaluation

This paper evaluates privacy with two kinds of attacks, namely, (a). distance based re-identification attack, and (b). inference attacks (attribute inference attack and membership inference attack).

**Re-identification Attack (RIA) :** The main idea behind RIA is, if an adversary has access to the the synthetic data-set (publicly open), can they re-identify an original record using distance based similarity? To simulate this attack, each of the synthetic record is picked and its closest match in the original record is found, using a distance based mechanism (any type of suitable distance metric can be utilized here [5]). If the corresponding distance is found to be lower than the specified cut-off then it is concluded that the adversary has successfully re-identified the real record. The privacy against RIA i.e. $P_R$ is quantified using the proportion of the samples for which the attack is not successful i.e.

'non-risky' samples in the data-set. A $P_R$ value of 90% means, 90% of the samples are not vulnerable to RIA. A high $P_R$ is desired.

**Attribute Inference Attack (AIA) :** Here, an adversary tries to reveal a particular sensitive field for a real record [4],[12]. This attack is modified based on the hypothesis of differential privacy. According to DP, the inclusion of a single record to the synthetic data-pool should not change the overall attack accuracy significantly. To simulate this attack, random original samples are attached with the synthetic data one at a time, to track the average change $e$ in AIA accuracy. The privacy against AIA is denoted with $P_A$. A $P_A$ value of 90% means, while revealing an attribute the probability of making error by the adversary is 90% (accuracy is 10%). A high $P_A$ is desired.

**Membership Inference Attack (MIA) :** With MIA, an adversary tries to infer if a synthetic sample can be matched to an original sample that belongs to the training data-set that was used to create the synthetic data-set in the first place [12] i.e. if its membership can be inferred. This can have significant impact. For example: If an adversary is aware that the synthetic data-set $S$ is cancer related and is also able to find out that a synthetic sample can be matched to the training set $R_1$, then they can infer that the victim has cancer. A $P_M$ value of 90% means, while trying to infer the membership the adversary is only 10% accurate (i.e. 90% error probability). A high $P_M$ is desired.

## 5  Analytical Utility Evaluation

The literature reports a significant dilution of analytical utility with stricter guarantee of privacy. In this paper, analytical utility is evaluated empirically using the following metrics,

**FID Score :**  Frechet Inception Distance or FID [13] is a measurement of similarity between two multivariate normal distributions which is used here to asses the utility of the data-set. Lower FID value indicates higher (closer to original) utility.

**JSD Score :**  Jensen-Shannon is a symmetric distance measure [14] for two uni-variate distributions. The utility can be quantified by first computing the JSD for each feature between original and the private data, and then taking an average over all features to get the final JSD score. Lower is better.

$\alpha$ **Precision ($\alpha$-P) :**  Here, the task is to estimate the likelihood of a private sample $y_j$ belonging to the real distribution $R$ within its $\alpha$-support (e.g. a random sub-set of the real data, determined by $\alpha$-mass). Higher value of $\alpha$ precision [13] indicates higher utility.

**Aggregated Correlation (Agg Corr):**  is the mean value of all the absolute bi-variate (pairwise) correlations. This is calculated from the upper triangular matrix of the cross-correlation table. This is used to estimate how well the correlations are retained in the private data. Higher is better.

**Machine Learning Utility (MLU) :**  Accuracy of a ML task trained on the private data and tested on real data is utilized here, to get an estimation for the ML utility. Higher MLU is expected.

Apart from the above metrics, a number of **Qualitative Analysis** are also performed to further explore the analytical utility of the private data such as: Uni-variate histogram (categorical)/ KDE (continuous) comparison, pairwise cross-correlation comparison and tSNE visualization. The detailed results for these analysis are moved to **supplementary materials** along with time and computational complexity, discussion on the attacks, results from other generative models, their hyper-parameters, data-set descriptions, implementation details and more.

## 6  Experimental Set-up

To illustrate the performance of PRISIM experiments are performed on multiple open-source data-sets. Subsequent subsections summarize the set-up and the obtained results.

**Data-set and Data Synthesizer** The efficacy of PRISIM against DP-CTGAN, is tested on four open-source tabular data-sets, Human Resource (HR) data-set captures employee attributes in an organization. ML task (for MLU and AIA) here is to predict monthly income. MIMIC-III (MIMIC) [15] captures Electronics Health Records (EHR) at patient level. BMI prediction is used for MLU and AIA. Airline satisfaction survey data (Airline) ML task is to predict the passenger satisfaction for MLU and AIA.Heart-failure data (Heart) has death event which is used as ML task for MLU and AIA. To show that the proposed DbP mechanism is agnostic of the underlying data synthesizer, three

| data-set | Data-type | Privacy | | | Utility | | | | |
|---|---|---|---|---|---|---|---|---|---|
| | | RIA (%) | AIA (%) | MIA (%) | FID | JSD | α-P | Agg Corr | MLU |
| **HR** | *Original* | - | - | - | - | - | - | 0.34 | 0.91 |
| | *Synthetic* | 51 | 67 | 62 | 0.02 | 0.22 | 0.42 | 0.33 | 0.88 |
| | *PRISIM* | 95 | **97** | **96** | 0.19 | 0.28 | 0.11 | 0.31 | 0.78 |
| | *DP-CTGAN* | **97** | **97** | **96** | 0.34 | 0.33 | 0.07 | 0.18 | 0.52 |
| **MIMIC** | *Original* | - | - | - | - | - | - | 0.27 | 0.99 |
| | *Synthetic* | 54 | 74 | 68 | 0.41 | 0.34 | 0.56 | 0.25 | 0.90 |
| | *PRISIM* | **98** | **96** | 94 | 0.53 | 0.36 | 0.09 | 0.21 | 0.86 |
| | *DP-CTGAN* | 97 | **96** | **95** | 0.72 | 0.42 | 0.04 | 0.11 | 0.56 |
| **Airline** | *Original* | - | - | - | - | - | - | 0.42 | 0.82 |
| | *Synthetic* | 57 | 78 | 72 | 0.15 | 0.41 | 0.66 | 0.25 | 0.81 |
| | *PRISIM* | 94 | 96 | 89 | 0.25 | 0.45 | 0.35 | 0.21 | 0.71 |
| | *DP-CTGAN* | **99** | **98** | **96** | 0.34 | 0.52 | 0.14 | 0.11 | 0.51 |
| **Heart** | *Original* | - | - | - | - | - | - | 0.32 | 0.79 |
| | *Synthetic* | 50 | 72 | 63 | 0.25 | 0.34 | 0.52 | 0.25 | 0.70 |
| | PRISIM | **97** | 94 | 92 | 0.32 | 0.36 | 0.29 | 0.19 | 0.65 |
| | *DP-CTGAN* | **97** | **98** | **96** | 0.38 | 0.42 | 0.21 | 0.08 | 0.45 |

Table 1: Results for different privacy attacks and utility metrics

types of generative models are explored, Conditional Tabular GAN (CTGAN), Tabular VAE, (TVAE), Gaussian Copula (GC). The overall findings (trends) stay similar hence for succinctness we will stick to the CTGAN (results for GC and TVAE are provided in supplementary materials).

**Privacy vs Analytical Utility Analysis** This section presents the numerical results and analysis for the experiments. Subsequently, the privacy vs utility trade-off is also discussed in detail. Based on Table 1 it is clear, that PRISIM is comparable with DP-CTGAN on all privacy metrics across all the data-sets. However, it is significantly better in terms of analytical utility (on an average by 22%).

Additional experiments are performed to compare the performance of PRISIM against state-of-the-art (SOTA) DP based generative models such as DP-WGAN[16], DP-VAE[17] and DP-auto-GAN [18]. These experiments are performed on the ADULT data-set. Aggregated JSD score (lower better) and ML Utility (salary prediction, higher better) are reported for the SOTA as analytical utility metrics, from [18]. Results from table 2 show the efficacy of the proposed mechanism in balancing both high privacy and analytical utility across different privacy budgets* [3] consistently.

| ADULT | Method | $\epsilon$ | JSD | MLU |
|---|---|---|---|---|
| | **DP-WGAN** | 0.36 | 1.29 | 0.75 |
| | | 0.50 | 2.41 | 0.76 |
| | | 1.00 | 0.73 | 0.77 |
| | **DP-VAE** | 0.36 | 0.8 | NA |
| | | 0.50 | 0.44 | NA |
| | | 1.00 | 0.23 | NA |
| | **DP-auto-GAN** | 0.36 | 0.33 | 0.74 |
| | | 0.50 | 0.23 | 0.78 |
| | | 1.00 | 0.19 | 0.79 |
| | **PRISIM** | $\sim 0.36*$ | 0.25 | 0.78 |
| | | $\sim 0.50*$ | 0.23 | 0.81 |
| | | $\sim 1.00*$ | **0.18** | **0.84** |

Table 2: Comparison of PRISIM with state-of-the-art DP based generative models

---

[3] A proxy $\epsilon$-privacy budget is approximated for DbP and matched with SOTA DP methods, in order to perform a fair comparison. For more details on this process please refer to the **supplementary material**

# 7 Conclusions

This paper presents a hybrid approach named PRISIM for generating private synthetic tabular data which integrates a distance based privacy mechanism with a deep learning based synthetic generation. PRISIM is extensively evaluated across multiple data-sets against state-of-the-art DP based privacy methods and it has consistently outperformed its peers in maintaining strong privacy while retaining better analytical utility. However PRISIM addresses privacy and utility concern only for tabular data-set so far. Therefore, further research is required to extend this approach to include other modalities such as time-series, images, text which also require privacy protection.

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
