# OpenReview forum: "PRISIM: Privacy Preserving Synthetic Data Simulator"
_NeurIPS.cc/2022/Workshop/SyntheticData4ML — Neurips 2022 SyntheticData4ML_

### Official Review · Reviewer_DaU9 · 2022-10-12
**Good synthetic data simulator for privacy preservation.**

**Rating:** 7
**Confidence:** 4

**Review:**

In this paper introduces PRISIM, a new method the generate data with privacy preservation. To define a distance between synthetic samples and real samples, the authors use Jaccard index for categorical variables and Mahalanobis distance for continuous variables. With the combined distance HMJD, the authors identify 'risky' synthetic samples. By removing risky samples, it gives a synthetic sample set with privacy preservation. The experiments show PRISIM is better than DP-methods.

## Strength

This paper is well-organized. The idea and motivation of PRISIM are very clear. The limitation of existing DP-methods is total noise grows rapidly when adding noise on gradient in each training step. It's an interesting idea think of other methods, as introduced in this paper by selecting desired synthetic samples instead of repeating adding noise. Though the authors use CTGAN as basic architecture, this idea of PRISIM can be implemented on other synthetic models.

The experiments are also convincing. As known, it's difficult to define a good synthetic sample set. The authors used 8 different metrics on 5 datasets to demonstrate the performance of PRISIM on different perspectives. And PRISIM outperformed other existing methods.

## Weakness

On one hand, I think it could be difficult to generate desired samples with fixed sample number. For example, just generate samples with same size of real samples. By keep removing 'risky' samples, I think it could be difficult to control how many samples are remained. If it is possible, the authors could give more explanation on that.

On the other hand, I think it could be better if the authors mention the time cost of their method. As mentioned in the paper, once removed 'risky' sample, re-evaluation is required. In some special cases, this method may require evaluation for many times. Then it could be very slow to get desired synthetic sample. So I think the authors could mention their training time cost in experiments or give some analysis on time complexity, then this work could be more convincing.

## Summary
Though there are some weakness to be refined, this work is great and interesting, with simple but clear idea and well-organized experiments.

---

### Official Review · Reviewer_zV8r · 2022-10-18
**The paper proposed distance based method to generate privacy preserving synthetic data. The literature review is solid. However,I have some concerns about the method and experiments**

**Rating:** 5
**Confidence:** 4

**Review:**

Quality: The literature review in main paper and supplementary is in great details.


clarity: The method is clear.


originality: I am not aware of similar method while distance based method has been proposed as cited in [5] original paper
significance: The innovation is incremental on top of Ads-Gan[5] (Better generative network and better distance metric)


experiment:
1. The paper does not state how many times it runs each method.
2. There is no other privacy preserving data generation method in Table 1. I would love to see its comparison with DP-Merf
3. How do you assign epsilon to your method? It does not have any differential privacy guarantee based on descriptions.
4. Why not compare your method with Ads-Gan type of method which is more similar than DP-VAE etc.

Method:
1. One of innovation is using distance metric HMJD. However, there is no comparison of CT-GAN with different distance metric. How do I know it performs the best?
2. If an attacker has access to the generated dataset, it may be easy for them to find few holes in the generated dataset. Those holes can be used to identify real data.

---

### Official Review · Reviewer_zFSH · 2022-10-19
**Review of "PRISIM: Privacy Preserving Synthetic Data Simulator"**

**Rating:** 6
**Confidence:** 3

**Review:**

This work focuses on the assessment and preservation of privacy and utility within centralized tabular data, and claims that the fidelity of synthetic samples from previous methods is generally poor and they do not provide empirical evidence of privacy. The paper proposes the PRISIM mechanism which combines CTGAN and distance based privacy, and provides empirical evidence of privacy and utility.

However, it is unclear whether PRISIM is time and computationally efficient and how many re-evaluation steps should be conducted in expectation.

---

### Meta-Review · Area_Chair_QaE5 · 2022-10-20

**Recommendation:** Accept